

# Perianth symmetry in sexually differentiated flowers of *Akebia quinata* (Lardizabalaceae)

Jiri Neustupa[1] and Katerina Woodard[2]

[1] Department of Botany, Faculty of Science, Charles University Prague, Prague, Czech Republic
[2] Czech Botanical Society, Prague, Czech Republic

## ABSTRACT

Sexual differentiation of monoecious plants usually involves differentiation in the size of female and male flowers produced on the same individuals. In the nectarless *Akebia quinata* (Houtt.) Decne (Lardizabalaceae), the trimeric, actinomorphic female flowers are larger than the males, which is explained as an adaptive trait to prevent self-pollination, as conspicuous female flowers are usually visited by pollinators earlier than smaller male flowers of the same individuals. This results in the plants being cross-pollinated rather than geitonogamously pollinated. However, it is also known that the development of the perianth in this species is genetically associated with the ontogeny of the petaloid sepals. These are thus developmentally linked to the ontogeny of the stamens. Therefore, it is possible that female flowers lacking fertile stamens also have less developmental control over the perianth ontogeny. Consequently, our study investigated whether female and male flowers of *A. quinata* differ in their overall shape features, in the amounts of variation among flowers, as well as in the extent of different types of asymmetry in perianth shapes. Geometric morphometric analyses of triradial perianth symmetry based on the generalised Procrustes analysis of a complete symmetry group of perianth shapes showed that female flowers were indeed significantly more variable in all different subspaces of their symmetric and asymmetric shape variation. This included the differences among individual flowers, their rotational and bilateral symmetry as well as the asymmetry among sepals within flowers. These results indicate that developmental control over perianth shape is systematically weaker in female flowers compared to male flowers of *A. quinata*. It is therefore likely that this phenomenon is related to the presence or absence of fertile male reproductive organs, whose development is linked to the ontogeny of the perianth and the maintenance of its trimeric symmetry.

## INTRODUCTION

Monoecious plants have separate male and female sexual organs in different flowers of the same plant. It is estimated that monoecy occurs in about 7% of flowering plants (*Renner, 2014*; *Karasawa, 2015*). In angiosperms, monoecious plant species have repeatedly evolved from ancestors with bisexual flowers, usually through either gynomonoecious or

Corresponding author
Jiri Neustupa,
neustupa@natur.cuni.cz

andromonoecious transitional stages (*Cronk, 2022*). From a developmental perspective, the separation of male and female function in different flowers may directly affect the ontogeny of the perianth due to the differential expression of genes associated with different sex organs.

In particular, the developmental link between the perianth and stamens is hypothesised by the ABCE model due to the function of B-class genes expressed during floral ontogeny of dicotyledonous plants (*Chanderbali et al., 2016*). Thus, the reduction of perianth size in male-sterile flowers, which occurs in most dioecious and gynodioecious species (*Delph, Galloway & Stanton, 1996*; *Eckhart, 1999*; *Delph, Touzet & Bailey, 2007*), has often been interpreted as a consequence of the reduced expression of these genes in the tepals (*Sobral & Costa, 2017*; *Zhang et al., 2022*). In addition, male-sterile flowers of two unrelated gynodioecious plants have been shown to exhibit reduced developmental control over perianth morphology, resulting in increased asymmetry of their corolla (*Neustupa, 2020*; *Neustupa & Woodard, 2021*). For these plants, it was therefore hypothesised that the replacement of stamens by staminodes and the subsequent lack of pollen production in purely pistillate flowers is related to perianth development *via* the function of B-class genes (*Chanderbali et al., 2016*) and thus responsible for the reduced morphological integration of the corolla, leading to higher shape asymmetry. This phenomenon was also observed in the bisexual *Geranium robertianum* L., where smaller flowers with relatively low pollen production exhibited greater asymmetry in corolla shape (*Frey & Bukoski, 2014*).

At the same time, plants with unisexual flowers often also experience evolutionary pressure through pollination by animal vectors. In dioecious or gynoecious species, the male flowers may be larger than the purely pistillate flowers due to greater selection pressure on pollen dispersal, leading to their enlargement and making them more attractive to pollinators (*Delph, Galloway & Stanton, 1996*). At the same time, it is known that pollinators tend to prefer flowers with a more precise perianth symmetry, even if they do not receive a reward, such as increased nectar production (*Møller & Eriksson, 1995*; *Møller & Sorci, 1998*). Thus, the morphological differentiation of perianth size and the shape symmetry in sexually differentiated flowers of the same species could be a kind of interplay between developmental and selective factors.

However, in plants where the sexes have separated in a way that results in monoecy or gynomonoecy, it is common for purely pistillate flowers to have a larger perianth than flowers providing the male function (*Delph, 1996*; *Kawagoe & Suzuki, 2002*). This is because the selection pressure to avoid self-pollination in a situation where flowers of both sexes are found on the same individuals can lead to an increase in the size of female flowers. These are then usually visited earlier by pollinators than the surrounding smaller male flowers of the same individual, resulting in the plants being outcrossed rather than geitonogamously pollinated (*Kawagoe & Suzuki, 2002*).

One of the well-documented examples of this phenomenon is *Akebia quinata* (Houtt.) Decne (Lardizabalaceae). It is a monoecious entomogamous vine species with a natural range in temperate East Asia (*Christenhusz, 2012*; *Christenhusz & Rix, 2012*). *A. quinata* forms dense lianaceous stands up to about 10 m tall, with deciduous or semi-evergreen pentameric leaves. Stems and branches are woody and they wind in a counter-clockwise

direction around the host plants or artificial substrates (*Christenhusz & Rix, 2012*). Thus, stands of this species typically form dense intermingled populations with vaguely distinguishable individuals.

The monoecious flowers of *A. quinata* are arranged in racemes with significantly larger female flowers located in the apical position and clusters of smaller male flowers at the base of racemes. The flowers are always actinomorphic, trimeric and they have a distinct perianth consisting exclusively of three petaloid sepals of pink to dark purple colour, formed in a single whorl (*Payne & Seago, 1968*; *Zhang & Ren, 2011*). It has been shown that petaloid morphology of sepals in the genus *Akebia* is related to the expression of B-class genes during their morphogenesis (*Shan et al., 2006*; *Carrive et al., 2020*). The flowers of *Akebia* are nectarless and, thus, the female flowers offer no reward to pollinators. However, due to their size, they serve as a visual attraction for insect pollinators, who visit them first and only then move on to the smaller male flowers (*Kawagoe & Suzuki, 2002*; *Kawagoe & Suzuki, 2003*). This mechanism limits geitonogamous pollination, which would otherwise significantly reduce the fruit production of the plants (*Kawagoe & Suzuki, 2005*; *Wang et al., 2022*). In addition to size differentiation, however, the question arises whether the male and female flowers also differ in the shape of their perianths and the degree of control over its trimeric symmetry. If it is true that the male function of flowers is inherently related to the maintenance of perianth symmetry, typically through a higher expression of B-class genes, then it should also be true for monoecious *A. quinata* that the shape of the perianth is significantly more symmetrical in male flowers than in female flowers. The main question of this geometric morphometric study can therefore also be framed as follows: do the perianths of female and male flowers in monoecious *A. quinata* differ in their shape and the degree of asymmetry associated with sexual differentiation of the flowers? To answer this question, we use the biological shape analysis of symmetry based on the geometric morphometrics of the perianth shapes of *A. quinata*, growing as a ruderal weed in the Czech Republic.

## MATERIAL & METHODS

### Sampling

The flowers of *A. quinata* were sampled on April 11 and 12, 2024, when the populations were in the main flowering phase (Figs. 1A–1D). All flowers originated from a single 7 m$^2$ wild population growing on a fence in the experimental garden of the Faculty of Science, Charles University Prague (50.072488N, 14.422923E) at 220 m above sea level. A total of 100 flowers (50 female, 50 male) were sampled at the time of anthesis. The flowers were immediately photographed in planar view at a fixed distance of 25 cm using a Canon EOS 1200D (Canon Inc., Oita, Japan) digital camera with an EFS 18–55 mm lens. To assess imaging error, each object was photographed twice.

### Digitization of the perianth forms

The perianth forms of *A. quinata* were digitised by a combination of six fixed landmarks (LMs) and three curves placed along the outline of each sepal (Figs. 1E, 1F). The fixed LMs were located at the points of contact between two adjacent sepals and at their tips,

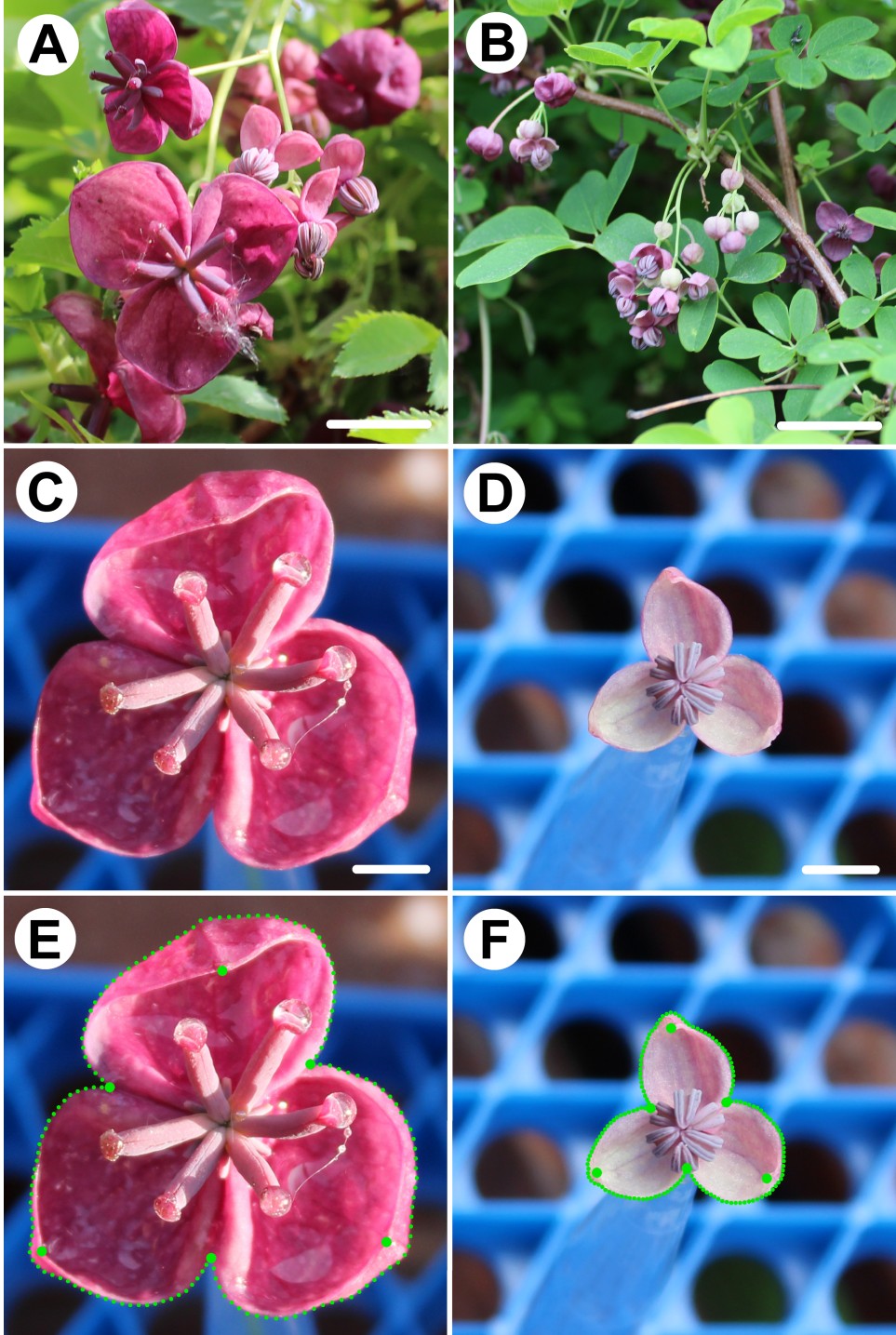

**Figure 1  Flowers of *Akebia quinata* and position of landmarks depicting the perianth forms.** (A) Flower raceme with prominent female flowers in distal position. (B) Flower raceme with male flowers. (C) Frontal view of a female flower. (D) Frontal view of a male flower. (E) Landmarks and semilandmarks depicting the forms of a female perianth. (F) Landmarks and semilandmarks depicting the forms of a male perianth. Scale bar = 10 mm (A, B), three mm (C, D).

which were usually hooded towards the centre of the perianth. Each of the three curves consisted of 50 equidistant points placed along the outline delimited by two fixed LMs at the visual intersection of adjacent sepals. Thus, these curves mapped the outlines of the sepals from their frontal view. In total, the entire configurations describing the form of each examined flower consisted of 156 points. To evaluate the digitisation error, all configurations were registered twice. The fixed LMs were digitised manually in TpsDig, ver. 2.22 (*Rohlf, 2015*), and the equidistant outline points were generated using the semi-automatic *draw_background_curve* function of this software.

## Geometric morphometrics

The geometric morphometric analysis of symmetry of the actinomorph perianth of *A. quinata* is based on six symmetry transformations that form the finite symmetry group of this structure. These transformations combine bilateral and rotational symmetry, which are typical for this type of purely actinomorphic flowers (*Savriama, 2018*). Thus, the dataset used in the generalised Procrustes analysis (GPA) consisted of six transformed copies of each individual flower (*Savriama & Klingenberg, 2011*; *Savriama, 2018*). These transformations corresponded to (1) identity, (2) rotation of the configurations by 120°, (3), rotation by 240°, and (4–6) bilateral reflections of the original and rotated configurations across the vertical axis of symmetry, accompanied by a corresponding relabeling of each corresponding landmark (Fig. 2). Consequently, the resulting dataset consisted of 100 (original flowers) × 2 (imaging) × 2 (digitisation) × 6 (symmetry transformations) = 2,400 landmark configurations. The size of each perianth was assessed using the measure of centroid size (CS), which is defined as the square root of the sum of the squared distances of all landmarks of an object from their centroid (*Bookstein, 2018*).

## Analysis of symmetric variation and measurement error

To investigate the effects of independent factors such as sexual differentiation, individual differences, repeated imaging, and digitization on symmetric components of shape variation, the configurations derived from the six symmetry transformations of each object were averaged (*Savriama, 2018*). This procedure resulted in perfectly symmetrical configurations of each of the studied flowers. Differences in shape variation among these configurations were assessed using a multivariate non-parametric type I Procrustes analysis of variance (ANOVA) (*Klingenberg, 2015*). The analysis decomposed the matrix of Procrustes distances among individual configurations into different sources specified by the independent factors. The significance of these factors was assessed by permutation tests based on the comparison of the Procrustes sums of squares (SS) spanned by the individual effects against the distribution of random SS resulting from 999 permutations (*Schaefer et al., 2006*). The permutation tests reflected the nested structure of the data. The fixed main effect "sex" was assessed by randomly redistributing the flowers between the male and female sex groups. Then, the effect of individual flowers was tested against the random SS distribution based on the reshuffling their multiple measurements yielded by repeated digitisations and imaging. The analysis was performed using the function *procD.lm* available in the package *geomorph*, version 4.0.8 (*Adams & Otárola-Castillo, 2013*), in R., version 4.4.1 (*R Development Core Team, 2024*).

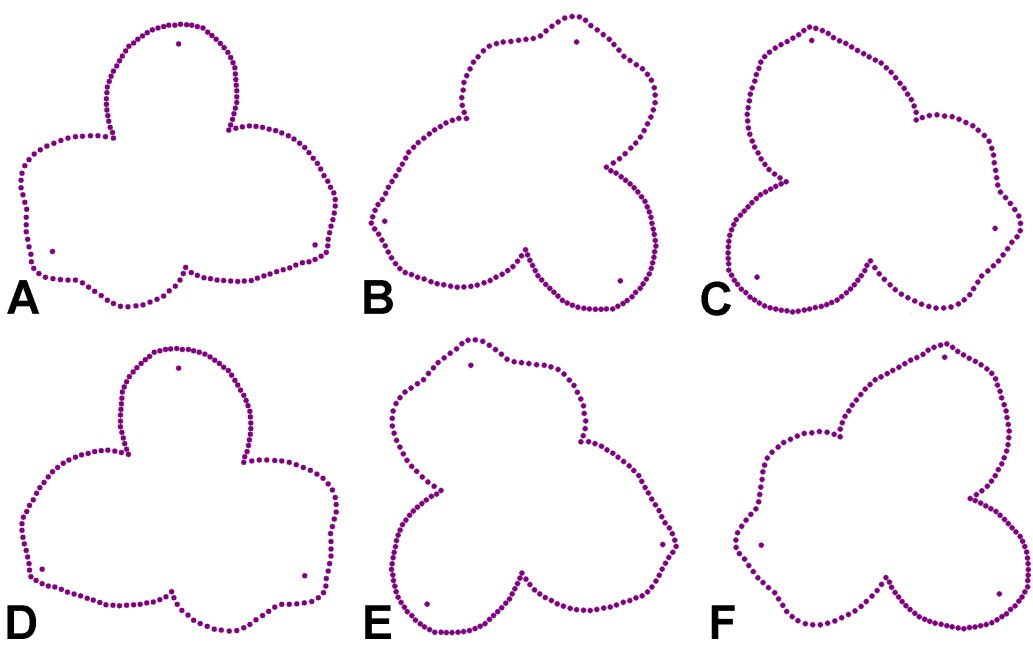

**Figure 2** **Symmetric transformations of a single perianth configuration of *A. quinata*.** (A) Original configuration. (B) Rotation by 120°. (C) Rotation by 240°. (D) Bilateral reflection of the original configuration. (E) Bilateral reflection of the configuration rotated by 120°. (F) Bilateral reflection of the configuration rotated by 240°.

## Analysis of shape symmetry

The decomposition of the individual components of symmetry and asymmetry in the perianth shape was based on a principal component analysis (PCA) of configurations representing a complete symmetry group of the studied data set. In such an analysis, individual principal components (PC) correspond to orthogonal subspaces separating different types of symmetry and asymmetry among the repeated parts of the analysed structure (*Savriama, Neustupa & Klingenberg, 2010*; *Savriama & Gerber, 2018*). In the case of a structure characterised by bilateral and rotational symmetry of order three, as in the case of *A. quinata*, there are subspaces corresponding to (1) total symmetry, in which the shape of all three parts changes in an identical manner, (2) rotational symmetry, characterised by an identical asymmetric change in each of the three sepals, (3) bilateral symmetry, which divides the entire perianth into two symmetric halves, and (4) total asymmetry, characterised by independent asymmetric changes in the shape of each sepal. Consequently, the PCs corresponding to total symmetry describe the differences among individual flowers in which the shapes of all three sepals change in a coordinated manner and remain mutually identical. The rotational symmetry corresponded to the pinwheel arrangement of the perianth, in which the individual sepals develop into coordinated asymmetric forms.

In contrast, the bilateral symmetry of the perianth disrupts the identical development of the three sepals because one of them is shaped differently from the other two, which are mirror-symmetrical to each other. Finally, total asymmetry shows the disintegration of

the perianth, with the individual sepals developing separately into different shapes along the axes belonging to this subspace of variation. It should be noted that these latter two types of shape variability occupy complementary subspaces, such that their eigenvalues and percentages of variability described are identical (*Savriama, 2018*). In a data set obtained by PCA, these PC pairs with the same eigenvalues form two-dimensional ordination spaces in which each configuration is represented by six points arranged symmetrically with respect to the three axes of symmetry. Thus, these PC pairs always represent a single unit describing the relevant part of the variability given by the sum of their eigenvalues and the two associated shape trends, one of which can be identified as the bilateral symmetry and the other as the overall asymmetry of the three flower parts.

The relative contributions of each subspace to the overall shape variation were quantified by summing the percentages of variation spanned by the PCs belonging to each of these subsets of perianth symmetry and asymmetry (*Savriama, 2018*). The amount of shape variation in each of these types was quantified as the sum of Euclidean distances on these PCs from their respective group centroids. It should be noted that for the asymmetric patterns, the group centroids are always in the centre of each PC. The 95% confidence intervals of the means were generated for the female and male flowers in each subspace of variation by bootstrap analyses with 9,999 replicates. In addition, the null hypothesis that the female and male flowers do not differ in the amounts of shape variation in each of the symmetrical and asymmetrical subspaces was evaluated by permutation tests based on 9,999 random replicates. These analyses were performed in PAST, version 4.10 (*Hammer, Harper & Ryan, 2001*). The thin plate splines representing the shapes typical of the marginal positions of each PC were reconstructed in TpsRelw, version 1.75 (*Rohlf, 2015*).

## RESULTS

Male and female flowers of *A. quinata* differed significantly in size. This difference was so pronounced that the distributions of each population sample did not overlap at all (Fig. 3). At the same time, the flowers of both sexes also differed in shape (Fig. 3). The individual sepals of the female flowers were broader and rounder, while the sepals of the male flowers were slightly elongated and pointed. The sepals of the female flowers were mostly bonnet-shaped in the distal part, so that landmarks denoting the position of their apex were clearly below the perianth outline (Figs. 1 and 3).

The significance of the shape differences between the two sexes was shown by the results of a multivariate Procrustes ANOVA (Table 1). The factor "sex" alone explained more than 54% of the total variation among flowers and it was highly significant with the lowest possible $p$-value = 0.001. Similarly, the differences between individual flowers within their sex groups were significantly non-random with respect to the two different sources of measurement error. Imaging error was slightly larger than digitising error, but both were found to be two orders of magnitude smaller than the effects examined. Therefore, the variation caused by these measurement errors could not affect the observed biological signal.

The combined PCA of the complete symmetry group of female and male flowers showed that of the two types of symmetry where all three sepals remained the same, total symmetry

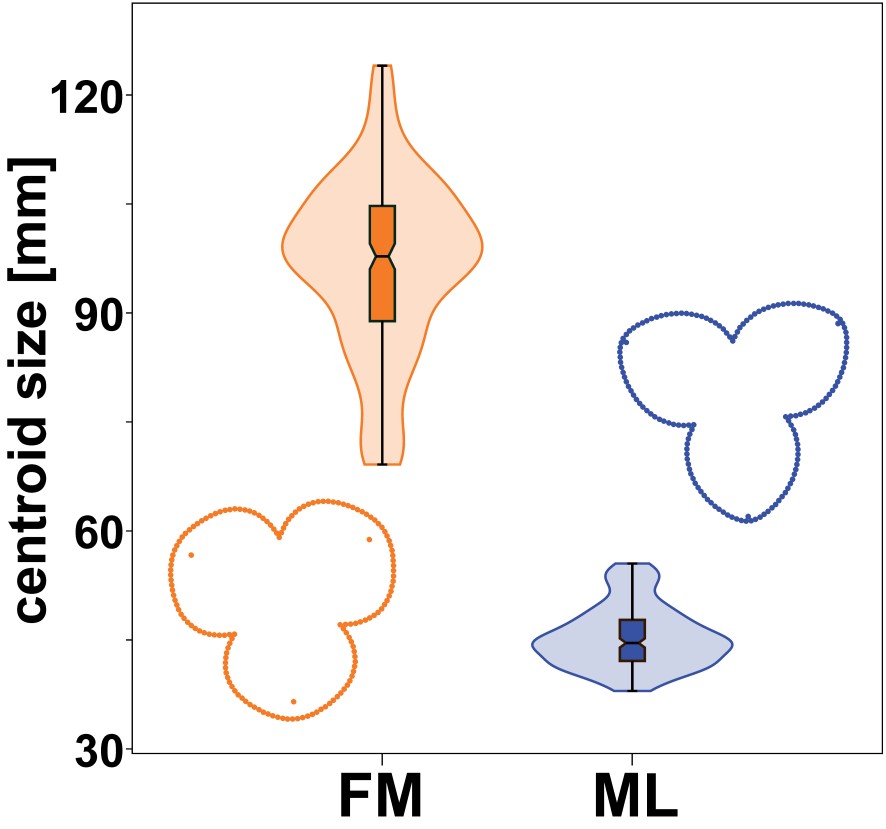

**Figure 3 Centroid size distribution of female (FM) and male (ML) flowers and the average perianth shapes.** The quantitative distribution of size values within each sex is indicated by the violin plots. Outlying values are indicated by whiskers on the edges of both box plots.

**Table 1 Results of multivariate analysis of variance evaluating variation among perianth shapes of *A. quinata* spanned by sexual differentiation, differences among flowers and measurement error.**

| Source | Df | SS | MS | $\eta^2$ | $p$ |
|---|---|---|---|---|---|
| Sex | 1 | 0.5624 | 0.56238 | 0.5439 | 0.001 |
| Flower (sex) | 98 | 0.4648 | 0.00474 | 0.4495 | 0.001 |
| Digitizing error | 100 | 0.0028 | 0.00003 | 0.0027 | |
| Imaging error | 100 | 0.0032 | 0.00003 | 0.0031 | |
| Digitizing: Imaging | 100 | 0.0008 | 0.00001 | 0.0008 | |
| Total | 399 | 1.0340 | | | |

**Notes.**
Df, degrees of freedom; SS, sum of squares; MS, mean squares; $\eta^2$, coefficient of determination; $p$, probability of the null hypothesis.

was significantly more prevalent than rotational symmetry (Fig. 4). Overall, the PCs describing total symmetry accounted for 32.57% of the shape variation, compared with only 2.15% in the case of rotational symmetry. PC1, which itself accounted for 25.34% of the total variation, belonged to the subspace of total symmetry with all six copies of
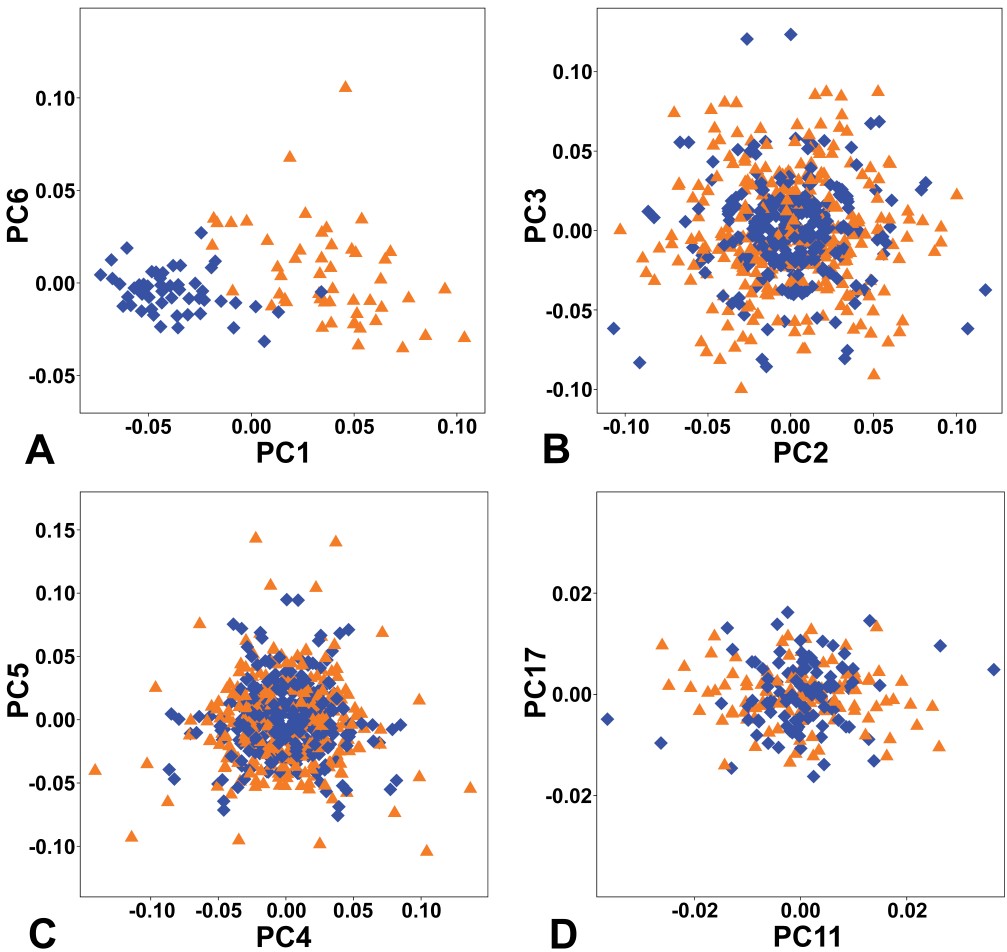

**Figure 4** **Ordination plots yielded by PCA of the complete symmetry group of perianth shapes showing two most important PCs from each subspace of symmetric and asymmetric variation.** (A) Totally symmetric variation. (B) Bilaterally symmetric variation. (C) Totally asymmetric variation. (D) Rotational symmetric variation. Blue diamonds represent the male flowers and orange triangles depict the female flowers, respectively.

each configuration having the same score on this PC and captured a shape trend between flowers with bluntly rounded sepals and those with narrower sepals and pointed apices, which seemed to primarily reflect their intersexual differentiation (Figs. 4A, 5A). Another axis that described a completely symmetrical shape dynamics was PC6 (5.49%), which emphasised the differences in the depth of incisions between adjacent sepals (Figs. 4A, 5F).

The first two axes belonging to the subspace of rotational symmetry were PC11 (1.27%) and PC17 (0.51%) (Fig. 4D). The six symmetrical transformations of one and the same flower always occupy two mirror positions on these axes, each represented by three configurations. These axes described shape differences in the bilateral asymmetry of individual sepals, but they were always mutually identical within the individual perianths. While PC11 described asymmetric changes mainly affecting the central parts of sepals,

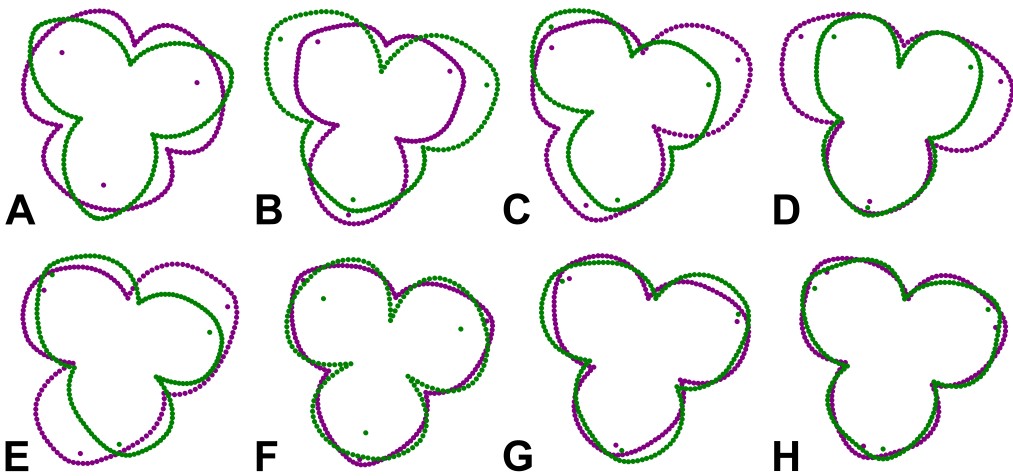

**Figure 5 Perianth shapes corresponding to marginal positions of individual PCs depicting the first two most important trends of variation in different subspaces of symmetric and asymmetric variation.** (A) PC1, total symmetry. (B) PC2, bilateral symmetry. (C) PC3, total asymmetry. (D) PC4, bilateral symmetry. (E) PC5, total asymmetry. (F) PC6, total symmetry. (G) PC11, rotational symmetry. (H) PC17, rotational symmetry. Shapes typical of opposite margins on each PC are depicted in violet and green colours.

PC17 described asymmetric changes in the shape of the apical parts of sepals, including their hooded tips (Figs. 5G, 5H).

The axes that combined bilateral symmetry and total asymmetry described a total of 32.64 * 2 = 65.28% of the total shape variation. The two most prominent pairs of these subspaces were reconstructed by PCA as PC2/3 and PC4/5 (Figs. 4B, 4C). The first pair described a totally asymmetric trend (Fig. 5C) in combination with bilateral symmetry, involving in particular a marked expansion of the two lateral sepals in combination with a shortening of the central sepal on the axis of bilateral symmetry (Fig. 5B). The other pair exhibited bilateral symmetry in addition to a completely asymmetrical development, which was described as a change in the degree of compression of the two lateral sepals, while the sepal located on the axis of symmetry remained more or less constant (Figs. 5D, 5E).

The variation between female and male flowers showed a consistent pattern across all subspaces of symmetry and asymmetry. Female flowers were always significantly more variable, both in terms of total symmetrical differences among flowers and in terms of asymmetry within flowers (Fig. 6). The confidence intervals of the mean variance values did not overlap between the two sexes in any of the four subspaces, and the permutation tests consistently showed highly significant differences (Table 2). The most pronounced difference between female and male flowers was found in symmetrical differences among flowers, with the variability of male flowers reaching only about 62% of the variability of female flowers. For asymmetrical subspaces, the variability of male flowers was about 75–76% of the observed asymmetrical variability of female flowers.

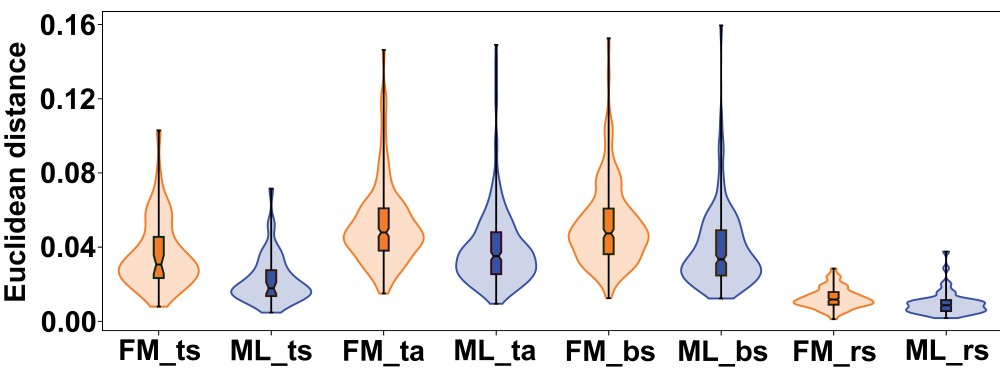

**Figure 6** **Violin plots showing the quantitative distribution of perianth shape variation in each of the two sexes and four subspaces of morphological symmetry and asymmetry.** FM, females; ML, males; ts, total symmetry; ta, total asymmetry, bs, bilateral symmetry, rs, rotational symmetry.

**Table 2** **Results of bootstrap tests for the difference in the amounts of shape variation in four subspaces of symmetry and asymmetry between the female (FM) and male (ML) flowers.**

| Symmetry/asymmetry subspace | Mean (FM) | SEM (FM) | 95 % CI for mean (FM) | Mean (ML) | SEM (ML) | 95 % CI for mean (ML) | *p* |
|---|---|---|---|---|---|---|---|
| Total symmetry | 0.0359 | 0.0026 | [0.0306, 0.0407] | 0.0223 | 0.0018 | [0.0185, 0.0257] | 0.0001 |
| Total asymmetry | 0.0519 | 0.0013 | [0.0495, 0.0543] | 0.0390 | 0.0012 | [0.0367, 0.0413] | 0.0001 |
| Bilateral symmetry | 0.0519 | 0.0013 | [0.0493, 0.0544] | 0.0390 | 0.0012 | [0.0365, 0.0413] | 0.0001 |
| Rotational symmetry | 0.0126 | 0.0003 | [0.0119, 0.0132] | 0.0097 | 0.0004 | [0.0090, 0.0104] | 0.0001 |

**Notes.**

CI, confidence interval; SEM, standard error of the mean; *p*, probability of equal means based on 999 permutations of the data.

## DISCUSSION

The analyses of shape symmetry in the perianths of *A. quinata* have shown that of the two symmetrical subspaces that do not disrupt the overall tripartite actinomorphy, total symmetry completely dominated over rotational symmetry. Thus, with a coordinated shape change of all three sepals, keeping the perfect actinomorphy of the flowers and precise within-flower developmental control over the ontogeny of individual perianth parts, it is relatively rare for this species to significantly change the flower shape towards the "pinwheel symmetry", as is typical for example for the genera *Vinca* L. or *Iris* L., where rotational symmetry plays a significant role in the composition of the symmetrical subspaces of shape variation within the actinomorphic perianths of these taxa (*Savriama, 2018*; *Tucić et al., 2018*).

However, asymmetric variation, disrupting regular actinomorphy of flowers, played an important role in the studied dataset. Decomposing a tripartite structure of this type does not allow us to distinguish the overall asymmetry of all three sepals from the tendency towards bilateral symmetry, but together these subspaces of variation accounted for almost two-thirds of the variation quantified by PCA. In these cases, then, we are dealing with a situation in which one or more sepals change their shape differently from the others within the same perianth. This type of deviation from an ideal actinomorphic arrangement of

flowers may be related to different and unrelated causes, such as an incipient evolutionary tendency towards zygomorphy (*Gómez & Perfectti, 2010*; *Berger et al., 2017*), an overall reduced developmental control during ontogeny of individual perianth parts (*Frey & Bukoski, 2014*), as well as to various external factors that act differently on different parts of the flower. For example, *Tucić et al. (2018)* and *Budečević et al. (2022)* showed that the effect of compass orientation of flowers leads to asymmetric development of the perianth structures due to the different irradiance of different flower parts. Although it is possible that such direction-giving factors also played a role at the level of individual flowers in our dataset (*Budečević et al., 2022*), it is implausible to presume that cardinal directions might be systematically different between the female and male flowers. It can therefore be assumed that the observed differences between the sexes are rather due to intrinsic factors acting during flower development.

Regarding the size differentiation of male and female flowers of *A. quinata*, our results clearly confirmed previous morphological descriptions (*Christenhusz & Rix, 2012*; *Wang et al., 2022*). The female flowers were on average slightly more than twice as large as the male flowers. In this respect, then, a monoecious, sexually differentiated entomogamous system such as *A. quinata* differs fundamentally from dioecious or gynodioecious sexually differentiated flowers, in which purely pistillate flowers of animal-pollinated taxa are usually characterised by their smaller perianths compared to flowers that fulfil the male function (*Delph, Galloway & Stanton, 1996*; *Barrett & Hough, 2013*). However, in this study we have also shown that this differentiation includes distinct differences in the shape of the perianth and individual sepals. Male flowers were also consistently more morphologically homogeneous, both in terms of differences between different flowers and asymmetric deviations from perfect triradial symmetry of the perianth within flowers. Here, our results are generally consistent with previous studies that analysed quantitative patterns of morphological symmetry in sexually differentiated flowers (*Frey & Bukoski, 2014*; *Neustupa, 2020*; *Neustupa & Woodard, 2021*). These studies have shown that flowers lacking the male function have reduced developmental control over the symmetry of the individual parts of their perianths. However, in gynodioecious and dioecious species, this was generally associated with a smaller size of the perianths of purely pistillate flowers, which were thus both smaller and less orderly developed compared to the flowers assuring the male function within their respective sexual system (*Delph, 1996*; *Neustupa, 2020*).

In the monoecious species *A. quinata*, however, the situation is different. The female flowers here are much larger, probably due to selection pressure on their visual attractiveness to attract insect pollinators before they settle on the male flowers in the same cluster, thus reducing the likelihood of geitonogamous pollination (*Kawagoe & Suzuki, 2002*; *Kawagoe & Suzuki, 2003*). Nevertheless, our analyses showed that the reduced symmetry of the perianths of female flowers was similar to previously studied gynodioecious species such as *Euonymus europaeus* or *Glechoma hederacea* (*Neustupa, 2020*; *Neustupa & Woodard, 2021*) or the bisexual *Geranium robertianum*, where the asymmetry of the corolla shape was also higher in smaller flowers with relatively low pollen production (*Frey & Bukoski, 2014*). Thus, our results suggest that the degree of developmental control over corolla symmetry is not necessarily allometrically related to flower size. However, they could indicate that

the male function of flowers is inherently linked to tightly kept perianth symmetry. Based on a classical Portmann framework that classifies the phenotypic expressions of living organisms, the perianth of *A. quinata* clearly functions as the so-called addressed biological phenomenon whose recipients are the pollinators (*Portmann, 1960*). The size of the perianth is here primarily determined by this interspecific relationship. In addition, however, the target phenotype is also influenced by intrinsic factors based on the developmental architecture of the flower. In general, the symmetry of the perianth appears to be distorted when flowers lack fertile stamens. Their presence correlates with a tightly controlled perianth symmetry, both in gynodioecious plants, in which the flowers providing the male function are larger, and in monoecious *A. quinata*, in which these flowers are significantly smaller.

In particular, the B-class genes are known for their characteristic joint expression in stamens and perianth (*Chanderbali et al., 2016*). While it has been shown that their increased expression in the perianth primarily leads to larger floral displays in comparison to the male-sterile flowers, it is possible that in monoecious species of the genus *Akebia* they are rather involved in the maintenance of the symmetry perianth symmetry. *Shan et al. (2006)* identified several B-class genes with different expression patterns in male and female flowers of *A. trifoliata*, a species closely related to *A. quinata*. For example, *AktAP3_2* gene was expressed at higher levels in male flowers than in female flowers and *AktPI* products were jointly found in the androecial primordia and the developing sepals (*Shan et al., 2006*). Thus, it is possible that these B-class genes are involved in tight control of perianth symmetry depending on the development of functional stamens.

However, it should be noted that our results were based on a local sample from a single population of the studied species. In addition, this is a population of a species growing outside its original distribution area that was anthropogenically introduced into the warmer areas of Central Europe (*Glasnović & FišerPečnikar, 2010*). Thus, we must be aware that the observed pattern of differential symmetry control over perianth shapes in female and male flowers may be variable across the range of *A. quinata*. In this context, it will therefore be important in the future to obtain more extensive morphometric data for other populations of species of the genus *Akebia* and other plant taxa with a monoecious reproductive system and size differentiation with relatively larger female flowers. If it is confirmed that smaller male flowers tend to have relatively more tightly kept perianth symmetry in such a system, *Bell*'s (*1985*) classical statement that "the flower is primarily a male organ" will still be valid in this respect. Biological shape analysis, such as geometric morphometrics of symmetry, provides suitable techniques for this research, which in the future could shed light on whether such general patterns of symmetry control actually exist in the floral development of angiosperms.

## CONCLUSIONS

Based on the biological shape analysis, this study has shown that the female and male flowers of the monoecious species *A. quinata* differ not only in size, but also in the shape of their triradial sepaloid perianths and the control of their symmetry. The male flowers are much

smaller, but their symmetry is more precise than that of the female flowers. The observed difference between female and male flowers was consistently reflected in all segments of symmetric and asymmetric shape variation of their trimeric perianths. We therefore hypothesise that precise developmental control of perianth symmetry is inextricably linked to the development of fertile male reproductive organs. Based on comparisons with other plant taxa with varying reproductive systems involving separation of their sexual organs into different flowers, we hypothesise that this may actually be a more general pattern of symmetry control in perianth development across angiosperms. However, it is clear that the general relevance of this phenomenon needs to be investigated in the future in a much larger number of different taxa from different plant lineages.

## ACKNOWLEDGEMENTS

The authors would like to thank Evelyn Woodard for her cooperation and support during the field work.

### Funding
The study was supported by institutional funds of Charles University Prague. The funders had no role in study design, data collection and analysis, decision to publish, or preparation of the manuscript.

### Grant Disclosures
The following grant information was disclosed by the authors:
Charles University Prague.

### Competing Interests
The authors declare there are no competing interests.

### Author Contributions
- Jiri Neustupa conceived and designed the experiments, performed the experiments, analyzed the data, prepared figures and/or tables, authored or reviewed drafts of the article, and approved the final draft.
- Katerina Woodard performed the experiments, prepared figures and/or tables, and approved the final draft.

### Data Availability
The primary data and the codes are available at Zenodo:

Neustupa, J., & Woodard, K. (2024). Supplementary data to "Perianth symmetry in sexually differentiated flowers of Akebia quinata (Lardizabalaceae)" [Data set]. Zenodo. Available at https://doi.org/10.5281/zenodo.15871315.

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
