# Peer review of "Perianth symmetry in sexually differentiated flowers of Akebia quinata (Lardizabalaceae)"

_PeerJ, doi:10.7717/peerj.20060_

## Round 0.1 · original submission · Major Revisions

· Academic Editor

Major Revisions

The authors should carefully consider the reviewers' comments, particularly those pertaining to the enhancement of the article's readability, and to revise the conclusions accordingly, as suggested by the reviewers.

**PeerJ Staff Note**: Please ensure that all review, editorial, and staff comments are addressed in a response letter and that any edits or clarifications mentioned in the letter are also inserted into the revised manuscript where appropriate.

Reviewer 1 ·

Basic reporting

Review of: “Perianth symmetry in sexually differentiated flowers of Akebia quinata (Lardizabalaceae)” [Note: this is a joint review.]

This study has investigated differences in size and symmetry between functionally male and female flowers of the monoecious plant Akebia quinata. A geometric morphometric analysis of floral symmetry is employed in this study to dissect shape variation (sampled from 100 flowers of two sexes) into different components to facilitate comparison between male and female flowers. The statistically sound results showed that in A. quinata, the smaller male flowers tend to be more symmetric with less variation in floral shape when compared to the bigger female flowers. This finding from this monoecious plant adds to our understanding of floral sexual dimorphic traits by complementing findings from dioecious plants in which a similar difference in floral symmetry has been shown to exist between male and female flowers, characterized by a reversed size difference. Together, these results suggest that floral symmetry is not necessarily allometrically related to flower size, and the developmental control of perianth symmetry may be linked to male functions.

Introduction:
In the Introduction, the authors argue that the morphological integration of the corolla could be mediated via the function of B-class genes, and differential expression of B-class genes between male and female flowers may lead to a difference in the degree of control over perianth symmetry. Specifically, it is mentioned (Lines 111-113) that higher expression of B-class genes could be responsible for perianth symmetry maintenance.

Major issue: (1) B-gene functional diversity should be discussed in more detail in relation to their hypothesis. Even though the expression of B-class genes has not been characterized in the species A. quinata, it has been characterized in another study (Shan et al. 2006), cited by the authors, for a closely related species, A. trifoliata. We suggest that the authors consider incorporating this data into their hypothesized relationship between symmetry control and B-class genes mediated male functions, or relate their findings to the expression data in the Discussion.

Minor issue: (1) In lines 99-100, the description of how male and female flowers are arranged on racemes is confusing. The authors refer to the position occupied by female flowers as the “top”; this appearance is likely caused by the orientation of the raceme due to the uneven weight distribution among flowers of the same raceme. However, judging from Figure 1A, the female flower occupies a basal position as it is the first to initiate, and the rest of the younger flowers are male. We suggest that the authors clarify this, either by using a description that explains the positions of flowers on a raceme or by explaining why the basal female flower appears to be on the top, to avoid confusion.

Experimental design

Methods section:
Major issues: No major problems.
Minor issue: (1) Add a reference for the statement “the identity of the individual sepals and thus the direction of their differentiation is not fixed among the different flowers” in Lines: 206-207.

Validity of the findings

Results section:
Major issues: none
Minor issues:
(1) We suggest adding a scree plot to show the percentage of variation accounted for by different PCs.
(2) Figure 5: include the meaning of the two colors used for the two perianth outlines in the caption.
(3) Table 2: Are p-values missing from rows 3-5?

Additional comments

Discussion section:
Major issues: none
Minor issues:
(1) Consider relating the idea of “an active effect of B-class genes on perianth symmetry ...” stated in Lines 341-343 to the B-class genes expression data of A. trifoliata, and the potential functional diversity of B-class gene homologs.
(2) In Line 135, the abbreviation for landmarks (LM) should be introduced here.
(3) Latin names not italicized in Lines: 46-47, 147, 354, 406.
(4) The manuscript could benefit from some further editing.

Reviewer 2 ·

Basic reporting

I consider geometric morphometric studies highly valuable for analyzing patterns of variation in size, shape, and asymmetry in biological groups. In the specific case of plants with male and female flowers, I consider it important to statistically demonstrate the differences between them to contribute to understanding their pollinator attraction mechanisms. This study uses geometric morphometrics to analyze the asymmetry of the sepals in Akebia quinata. The manuscript is well written, and the figures are of good quality, so it has the potential for publication in PeerJ after making adjustments to its structure.

I would like to highlight two points that somewhat weaken the original version of the manuscript, in which the authors could make improvements.

1. I don't know the floral variation of A. quinata throughout its distribution (at least as a native species), but it seems very limited to discuss patterns of asymmetry in the species with the sample used. The authors mention "all flowers originated from a single 7 m2 wild population," but the number of sampled individuals is not described, which should be mentioned. I understand that the pattern was basically analyzed in one or a few individuals; therefore, the results reflect very specific variation in the sampled individual(s). Therefore, the authors should clarify the limitations of this sampling in their discussion and be more careful with their hypotheses.

2. I perceive that some parts of the paper attempt to associate morphological patterns with the expression of class B genes (abstract, discussion). However, I believe the sample was too small to make such assertions. I believe the authors can maintain this idea in the discussion, but point out the sample limitations of the paper.

Other specific recommendations:
Abstract: Lines 29-41: I suggest shortening the first phrases and briefly adding the analyses performed.

I hope these recommendations will help improve the manuscript.

Experimental design

Materials & Methods (M&M): I recommend that the M&M specifically describe the methods used. Some phrases in the "Analysis of shape symmetry" section would be more appropriate for the results and/or discussion.

Line 123: I suggest adding a brief topic with a description of the species to give the reader some context about plant morphology. Authors may cite Fig. 1(A, B) in this section.
Line 126: Add the number of sampled individuals.
Line 139: Add the meaning of LM before the abbreviation. Do this for each abbreviation.
Lines 196-197: Personally, I don't consider it necessary to include these sentences in M&M, since their methods do not analyze ontogenetic development in the species, and the analysis of asymmetry can be addressed in the discussion. If the authors wish, they can revisit the idea in the discussion.

Validity of the findings

Discussion: Line 301: It is insecure to make extrapolations and hypotheses about the genetic cause of shape and asymmetry in A. quinata, which did not include a larger number of individuals from different populations of the species.

301, 309-310, 329, 341-343: Provide references.

Reviewer 3 ·

Basic reporting

The article is written in professional, clear, and unambiguous English. The references used to support the theoretical framework, materials, and methods, and the discussion are broadly sufficient and comprehensive. It is important to highlight that the authors made a particular effort in formatting the references according to the journal’s guidelines. The manuscript follows a clear and professional structure, in line with PeerJ standards. The figures are relevant and of high quality. Although the authors made an effort to clarify the titles and legends of the figures and tables, these are insufficient, particularly in Figure 3 and Table 1. All legends lack critical information (description and conclusions) regarding the main findings each figure intends to communicate. This omission makes the figures heavily dependent on the main text and prevents them from being understood in isolation. It must be remembered that a figure and its legend should be fully interpretable independently, allowing comprehension, interpretation, and conclusions to be drawn without referring back to the main text.

Experimental design

The article aligns fully with the Aims and Scope of PeerJ: the subject area of developmental biology and plant morphology falls within the journal’s scope. It presents an original research article, consistent with the types accepted by the editorial board, and follows methodological criteria of rigour and reproducibility. The research question is clearly stated at the end of the introduction, along with its relevance and rationale. From the beginning, the manuscript explains how this research addresses a gap in scientific knowledge. According to the description, the study is conducted rigorously, adhering to high standards of technology and ethics. The authors have made a clear and thorough effort to describe and justify the methodology and how the data were analysed. Unfortunately, this may confuse readers, as the section contains interpretative elements that cross the boundary between “Materials and Methods” and the “Discussion.”

Validity of the findings

All underlying data have been generated for this study. The data are solid, statistically relevant, and well-controlled. However, the sampling was carried out within a single 7 m² area and from a single population. The target species is a liana that can grow up to 10 metres in length, meaning it is highly likely that the flowers came from very few individuals. This raises a risk of pseudoreplication, which may undermine the generalisations the authors wish to make. This limitation should be clearly acknowledged somewhere in the manuscript—ideally in the Discussion. The study repeatedly emphasises that flower size determines pollinator behaviour (this is cited in the Introduction, Discussion, and Conclusions). While this idea can serve as a conceptual background, it should be noted that the study’s actual objective and scope are strictly morphometric. The conclusion section is disproportionately long relative to the rest of the manuscript. It includes references, theoretical content, and evolutionary concepts that should be part of the Discussion. I suggest that the conclusions be more concise and focused on highlighting the novel contributions of the study, how it differs from previous work, and which theoretical gaps it helps to address.

Additional comments

The manuscript presents a very interesting and comprehensive morphological study. It employs several precise and well-documented methodologies. The manuscript is well written and well structured. Only a few minor typographical issues were found, along with five more substantive points that do not invalidate the study but should be addressed for publication in PeerJ. These are my recommendations:

a. The authors have made a special effort to describe and justify their methodology and data analysis clearly. However, this section includes excessive interpretation that blurs the line between “Materials and Methods” and the “Discussion.”

b. Sampling was conducted in a single 7 m² area, likely involving very few individuals, given that the species is a liana that may reach 10 metres in length. This introduces a strong risk of pseudoreplication, which may compromise the generalisability of the results. This limitation should be acknowledged in the manuscript, preferably in the Discussion.

c. The manuscript strongly emphasises that flower size determines pollinator behaviour (referenced in the Introduction, Discussion, and Conclusions). While this idea can be cited as part of the conceptual framework, it is important to reiterate that the scope of this study is strictly morphometric.

d. The conclusion section is disproportionately long. It introduces new references, theoretical content, and evolutionary concepts that should instead be discussed in the main body of the Discussion. I suggest the conclusions be made more concise and focused on the novel insights provided by this study, what sets it apart from similar studies, and the theoretical gaps it helps to fill.

e. Although the authors have made an effort to clarify figure and table titles and legends, they remain insufficient, especially Figure 3 and Table 1. All legends lack key information (descriptions and conclusions) about the main findings each figure or table aims to show. As a result, the legends are overly dependent on the main text. It is important to remember that a figure and its legend should be sufficient on their own for full interpretation and understanding.

Line-specific comments:
1. L.66: “2022a” – There is only one publication cited by these authors for that year. The “a” is unnecessary.
2. L.95–98: “However… 2010” – Due to the sampling limitations, this statement may be misleading at this point in the manuscript (Introduction). I suggest moving it to the Discussion, where the actual genetic variability of the population can be discussed.
3. L.128–129: “A total of 100 flowers… anthesis” – How many individual plants did these flowers come from?
4. L.135: Add “(LM)” after the word “landmarks.”
5. L.250: Cite Figure 4 immediately after “symmetry.”
6. L.354: A. quinata should be in italics.
7. L.386: Remove the period before the DOI.
8. L.447: Add a colon after the DOI.
9. L.476: Remove the period before the DOI.
10. L.494: Remove the period before the DOI.

---

## Round 0.2 · accepted · Accept

· Academic Editor

Accept

The authors addressed the reviewers' comments. The revised manuscript now meets the requirements for publication.

Reviewer 3 ·

Basic reporting

All previous remarks have been thoroughly addressed, and the revised manuscript meets the requirements for publication.

Experimental design

All previous remarks have been thoroughly addressed, and the revised manuscript meets the requirements for publication.

Validity of the findings

All previous remarks have been thoroughly addressed, and the revised manuscript meets the requirements for publication.

Additional comments

All previous remarks have been thoroughly addressed, and the revised manuscript meets the requirements for publication.